# Impact of early detection on cancer curability: A modified Delphi panel study

**Lee Schwartzberg[1‡], Michael S. Broder[2ⓒ], Sikander Ailawadhi[3‡], Himisha Beltran[4‡], L. Johnetta Blakely[5‡], G. Thomas Budd[6‡], Laurie Carr[7‡], Michael Cecchini[8‡], Patrick Cobb[9‡], Anuraag Kansal[10ⓒ], Ashley Kim[10ⓒ]\*, Bradley J. Monk[11‡], Deborah J. Wong[12‡], Cynthia Campos[2ⓒ], Irina Yermilov[2ⓒ]**

1 Division of Medical Oncology and Hematology, Renown Institute for Cancer, Reno, Nevada, United States of America, 2 Partnership for Health Analytic Research (PHAR), LLC, Beverly Hills, California, United States of America, 3 Department of Medicine, Division of Hematology/Oncology, Mayo Clinic, Jacksonville, Florida, United States of America, 4 Department of Medical Oncology, Divisions of Genitourinary Oncology and Molecular and Cellular Oncology, Dana Farber Cancer Institute, Boston, Massachusetts, United States of America, 5 Health Economics and Outcomes Research, Tennessee Oncology, Nashville, Tennessee, United States of America, 6 Department of Hematology and Medical Oncology, Cleveland Clinic Taussig Cancer Institute, Cleveland, Ohio, United States of America, 7 Department of Medicine, Division of Medical Oncology, National Jewish Health, Denver, Colorado, United States of America, 8 Department of Internal Medicine, Division of Medical Oncology, Yale University School of Medicine, New Haven, Connecticut, United States of America, 9 Oncology Research, Intermountain Healthcare, Billings, Montana, United States of America, 10 Health Economics and Outcomes Research, GRAIL, LLC, a subsidiary of Illumina Inc., currently held separate from Illumina Inc. under the terms of the Interim Measures Order of the European Commission dated 29 October 2021, Menlo Park, California, United States of America, 11 Department of Obstetrics and Gynecology, Division of Gynecologic Oncology, HonorHealth Research Institute, University of Arizona, Creighton University, Phoenix, Arizona, United States of America, 12 Department of Medicine, Division of Hematology/Oncology, UCLA Health, Los Angeles, California, United States of America

ⓒ These authors contributed equally to this work.
‡ LS, SA, HB, LJB, GTB, LC, MC, PC, BJM and DJW also contributed equally to this work.
\* akim@grailbio.com

**Data Availability Statement:** All relevant data are within the article.

**Funding:** This study was sponsored by GRAIL, LLC, a subsidiary of Illumina, Inc., currently held separate from Illumina, Inc. under the terms of the

## Abstract

Expert consensus on the potential benefits of early cancer detection does not exist for most cancer types. We convened 10 practicing oncologists using a RAND/UCLA modified Delphi panel to evaluate which of 20 solid tumors, representing >40 American Joint Committee on Cancer (AJCC)-identified cancer types and 80% of total cancer incidence, would receive potential clinical benefits from early detection. Pre-meeting, experts estimated how long cancers take to progress and rated the current curability and benefit (improvement in curability) of an annual hypothetical multi-cancer screening blood test. Post-meeting, experts rerated all questions. Cancers had varying estimates of the potential benefit of early cancer detection depending on estimates of their curability and progression by stage. Cancers rated as progressing quickly and being curable in earlier stages (stomach, esophagus, lung, urothelial tract, melanoma, ovary, sarcoma, bladder, cervix, breast, colon/rectum, kidney, uterus, anus, head and neck) were estimated to be most likely to benefit from a hypothetical screening blood test. Cancer types rated as progressing quickly but having comparatively lower cure rates in earlier stages (liver/intrahepatic bile duct, gallbladder, pancreas) were estimated to have medium likelihood of benefit from a hypothetical screening blood test. Cancer types rated as progressing more slowly and having higher curability regardless of stage (prostate, thyroid) were estimated to have limited likelihood of benefit from a

Interim Measures Order of the European Commission dated 29 October 2021. The funding sponsor had no role in the collection, management, and analysis of the data. The sponsor contributed to data interpretation. SA: Dr. Ailawadhi reported institutional research grants from Pharmacyclics, Amgen, Janssen, Ascentage, Xencor, Cellectar, BMS; consulting fees from BMS, Takeda, Oncopeptides; participating on a Data Safety Monitoring Board or advisory board for GSK, Sanofi. HB: Dr. Beltran reported institutional research grants from Janssen Oncology, BMS; consulting fees and advisory board participation for Janssen, Astellas, AstraZeneca, Merck, Pfizer, Foundation Medicine, Blue Earth Diagnostics, Amgen. JB: Dr. Blakely reported consulting fees from GRAIL, LLC, a subsidiary of Illumina, Inc., currently held separate from Illumina, Inc. under the terms of the Interim Measures Order of the European Commission dated 29 October 2021. GTB: Dr. Budd reported institutional research grants from Daiichi, Roche, Ambrx, Tracon, Ayala, Salarius; consulting fees from Roche, Partnership for Health Analytic Research (PHAR), LLC, GRAIL, LLC, a subsidiary of Illumina, Inc., currently held separate from Illumina, Inc. under the terms of the Interim Measures Order of the European Commission dated 29 October 2021. LC: Dr. Carr reported consulting fees from Takeda, Sereno EMD, AstraZeneca, Blueprint Medicine, Eli Lilly; payment for expert testimony from Kline & Spector P.C. and Osteen & Harrison, PLC. MC: Dr. Cecchini reported a grant from the National Cancer Institute Care Development Award; consulting fees from Agios Pharmaceuticals, Eisai Inc, AstraZeneca; support for attending meetings and/or travel from Genentech, Orche; stock or stock options from Parthenon Therapeutics. PC: Dr. Cobb reported consulting fees from Partnership for Health Analytic Research (PHAR), LLC. ARK: Dr. Kansal reported stock or stock options in Illumina (parent company of GRAIL, LLC, a subsidiary of Illumina, Inc., currently held separate from Illumina, Inc. under the terms of the Interim Measures Order of the European Commission dated 29 October 2021. Dr. Kansal is an employee of GRAIL, LLC, a subsidiary of Illumina, Inc., currently held separate from Illumina, Inc. under the terms of the Interim Measures Order of the European Commission dated 29 October 2021., who supported this study. AK: Dr. Kim reported stock or stock options in Illumina (parent company of GRAIL, LLC, a subsidiary of Illumina, Inc., currently held separate from Illumina, Inc. under the terms of the Interim Measures Order of the European Commission dated 29 October 2021). Dr. Kim is an employee of GRAIL, LLC, a subsidiary of Illumina, Inc., currently

hypothetical screening blood test. The panel concluded most solid tumors have a likelihood of benefit from early detection. Even among difficult-to-treat cancers (e.g., pancreas, liver/intrahepatic bile duct, gallbladder), early-stage detection was believed to be beneficial. Based on the panel consensus, broad coverage of cancers by screening blood tests would deliver the greatest potential benefits to patients.

## Introduction

In 2022, an estimated 1.9 million people will be diagnosed with cancer in the United States (US) [1]. Cancer is the second leading cause of death in the US, resulting in more than 600,000 deaths in 2020 [2].

Early detection of cancer is an accepted critical component of prevention and reducing cancer-related mortality. The main goal of screening is to identify cancers, or cancer precursors, early to reduce mortality. Existing cancer screening tests include imaging (e.g., digital mammography), serial exams (e.g., colonoscopy), and tissue sampling (e.g., Pap test), with serum markers (e.g., prostate-specific antigen [PSA] test) measured on an individual basis. Newer approaches to early cancer detection include blood-based screening tests, including multi-cancer tests, which may potentially be used to screen for multiple cancer types simultaneously [3].

Cancer screening reduces cancer mortality by detecting disease at an earlier stage when interventions may be more successful. For example, a US Preventive Services Task Force (USPSTF) meta-analysis showed a 15% to 20% reduction in breast cancer mortality with mammography screening and a 20% to 60% reduction in cervical cancer mortality with cytology-based screening [4–6]. In addition, the International Agency for Research on Cancer (IARC) reported a 22% to 31% reduction in colorectal cancer mortality associated with sigmoidoscopy screening [7]. Using Surveillance, Epidemiology and End Results (SEER) data, Clarke et al. estimated that detecting all cancer at stage III rather than stage IV would reduce cancer deaths by 15%, with larger gains if those cancers could be detected in even earlier stages [8].

Not all screening tests will decrease mortality, however. According to SEER data on thyroid cancer, kidney cancer, and melanoma, there has been an increase in rates of new diagnoses but not deaths. This suggests that these new diagnoses may include cancers that are less likely to be fatal and that screening for these cancers may result in overdiagnosis [9, 10], or an increase in detection of cancer incidence without a comparable reduction in late-stage disease or mortality [11]. In addition, screening may result in an early cancer diagnosis, generating a lead time bias in outcomes and artificially inflating the length of time a patient is considered to have cancer [12].

The benefits of early detection may vary across cancers. Screening programs have been particularly effective in colon [7] and cervical [13] cancers where precursor lesions are identified and removed; however, many types of cancer exhibit a range of heterogeneous behaviors and variable likelihoods of progression and death [11]. For other cancers, it is difficult to identify the at-risk population. In anal cancer, 91% of cases occur in patients with no perceived risk of invasive anal cancer, suggesting most anal cancers would be missed if only high-risk individuals were screened [14].

Given these complexities, we elicited expert input from practicing oncologists to understand which cancer types may benefit most from early diagnosis and what drove differences between cancer types.

held separate from Illumina, Inc. under the terms of the Interim Measures Order of the European Commission dated 29 October 2021., who supported this study. BJM: Dr. Monk reported consulting fees from Abbvie Inc, Agenus, Akeso Bio, Aravive, AstraZeneca, Clovis, Eisai, Elevar, EMD Merck, EMD Serono, Inc, Genmab/Seattle Genetics, GOG Foundation, Gradalis, ImmunoGen, Incyte, Iovance, Janssen Research & Development, LLC, Karyopharm, Merck, Mersana, Myriad, Novocure, Pfizer, Regeneron, Roche/Genentech, Sorrento, Takeda, TESARO/GSK, US Oncology Research, VBL; payment or honoraria from AstraZeneca, Clovis, Eisai, Merck, Roche/ Genentech, TESARO/GSK. DJW: Dr. Wong reported institutional research grants from Lilly, Merck Sharp & Dohme, Pfizer, TopAlliance, FSTAR, BMS, KURA, CheckMate, Roche/Genentech, Iovance, AstraZeneca, Elevar Therapeutics, LOXO Oncology, Astellas, Enzychem; consulting fees from Sanofi, Blueprint Medicine, Regeneron. MSB, IY, CC: Dr. Broder, Dr. Yermilov, and Ms. Campos are employees of Partnership for Health Analytic Research (PHAR), LLC which was paid by the following to conduct research related to the work described in the manuscript: Abbvie, Amgen, AstraZeneca, Biomarin Pharmaceuticals, BMS, Celgene, Dompe, Eisai, Exact Sciences Corporation, Genentech, GRAIL, LLC, a subsidiary of Illumina, Inc., currently held separate from Illumina, Inc. under the terms of the Interim Measures Order of the European Commission dated 29 October 2021., Ionis, Jazz, Kite, Novartis, Otsuka, Recordati, Regeneron, Sanofi US Services, Takeda Pharmaceuticals USA.

**Competing interests:** I have read the journal's policy and the authors of this manuscript have the following competing interests: LS, SA, HB, JB, GTB, LC, MC, PC, BJM, DJW reported consulting fees from Partnership for Health Analytic Research (PHAR), LLC and GRAIL, LLC, a subsidiary of Illumina, Inc., currently held separate from Illumina, Inc. under the terms of the Interim Measures Order of the European Commission dated 29 October 2021. ARK and AK are employees of GRAIL, LLC, a subsidiary of Illumina, Inc., currently held separate from Illumina, Inc. under the terms of the Interim Measures Order of the European Commission dated 29 October 2021, who supported this study, and reported stock or stock options in Illumina (parent company of GRAIL, LLC, a subsidiary of Illumina, Inc., currently held separate from Illumina, Inc. under the terms of the Interim Measures Order of the European Commission dated 29 October 2021). MSB, IY, and CC are employees of Partnership for Health Analytic Research (PHAR), LLC which was paid by

## Methods

We used the RAND/University of California, Los Angeles (UCLA) modified Delphi panel method, which is fully described elsewhere [15–17]. This method is a formal group consensus process which systematically and quantitatively combines expert opinion and systematic literature review evidence by asking panelists to rate, discuss, and then rerate various patient scenarios. Our panel included 10 experts, which falls within the recommended panel size of 7–15 which permits sufficient diversity and ensures a chance to participate, as per the RAND/UCLA Appropriateness Method guidelines [17]. The criteria to recruit oncology experts included having a breadth and diversity of oncology experience and representing different geographic regions and practice settings in the US. Panelists had an average of 20 years in clinical practice from a variety of practice settings (six academic, three community, and one combined academic and community) and US regions (three from the South, four from the West, two from the Northeast, and one from the Midwest). Panelists included a general practice oncologist and a diversity of oncology subspecialties, with expertise covering the range of cancer types considered. The number of experts with experience for specific cancer types were as follows: hematologic (2), prostate (1), breast (3), lung (2), colorectal (2), gastrointestinal (1), head and neck (1), liver (1), gynecological (1), and sarcoma (2). Expert panelists gave written informed consent for research participation and received honoraria from the study sponsor for their participation. Modified Delphi panels do not involve human subjects as defined by 45 Code of Federal Regulations part 46 and therefore do not require Institutional Review Board approval.

Prior to the meeting, we collaboratively developed a detailed, 540-item written questionnaire, or rating form, through individual phone interviews. We designed the rating form to obtain expert input regarding which cancers may benefit most from early detection (e.g., using a hypothetical multi-cancer screening blood test) and whether there are cancers for which the treatment outcomes would not change, if detected early.

The panel assessed whether a given cancer was considered "curable" at each stage, and how quickly it progresses from the beginning of one stage to the beginning of the next. In addition, we asked panelists to consider the risk of overdiagnosis. We defined "cure" as the receipt of effective treatment such that a population of individuals who are "cured" have the same life expectancy as a population that never had the cancer being considered. Curability was rated on a scale of 1 (extremely unlikely to be cured) to 9 (extremely likely to be cured). Cancer progression was rated on a scale of 1 (less than a year) to 9 (nine or more years). For both questions, experts referred to 10-year survival data from SEER as a proxy for cure and progression. As such, the questions on the rating form were designed and ordered specifically around this fundamental question of curability and developed in an iterative process in conjunction with the panelists to appropriately discuss the benefits and harms of diagnosing cancer at earlier stages. Scores were collected for all cancer types from all experts.

Experts were asked to estimate the benefit of an annual hypothetical screening blood test that is 100% sensitive and 100% specific for patients aged 50 years and older. They were told to assume that the blood test could not detect premalignant tumors and was not meant to replace any existing cancer screening tests. This hypothetical scenario was chosen to focus on the potential benefits of early detection in each cancer type, rather than to assess the impact of any specific screening technology or program. The benefit was defined as the likelihood that cure rates would increase on a scale of 1 (not at all likely to increase) to 9 (likely to increase a great deal).

Experts were also asked to consider both typical treatment (i.e., the care provided to the population as a whole) and best available care (i.e., care consistent with the National Comprehensive Cancer Network [NCCN] guidelines). Although the goal of treatment is to provide the best available treatment to all patients, there is evidence this does not always occur [18–20].

Experts considered curability across stages I to IV of 20 solid organ cancers, representing >40 American Joint Committee on Cancer (AJCC)-identified cancer types and 80% of total cancer incidence. Cancer subtypes were not considered. They completed ratings before a panel meeting in December 2020 (S1 Table). During the meeting, experts were provided with their first-round individual ratings and the panel's median ratings for all questions. As is standard in the RAND/UCLA modified Delphi panel method, we defined panel disagreement as at least 2 ratings of 1 to 3 and at least 2 ratings of 7 to 9 [17]. During the professionally moderated group discussion, panelists shared reasons for their ratings, focusing on areas of disagreement. After the meeting, panelists rerated all questions (S2 Table). Statements describing the group consensus that emerged from the second-round ratings were developed and circulated to all experts for review and approval.

## Results

After a group discussion, panelists disagreed on 1% of the 540 ratings, compared to 13% disagreement after the first-round ratings.

### Cancer curability and progression

The ratings of curability are provided in Table 1. Experts rated 85% (n = 17) of cancers as somewhat likely to extremely likely to be cured in stage I, 60% (n = 12) in stage II, 5% (n = 1)

**Table 1. Median (range) rating scores of the likelihood of cancer curability today.**

| Cancer type | Stage I | Stage II | Stage III | Stage IV |
|---|---|---|---|---|
| Thyroid | 9 (8–9) | 8 (7–9) | 7 (6–9) | 5 (1–7) |
| Colon/rectum | 9 (8–9) | 8 (7–8) | 5 (3–6) | 1.5 (1–3) |
| Kidney | 9 (7–9) | 8 (6–8) | 5 (4–6) | 2 (1–2) |
| Uterus | 9 (8–9) | 8 (7–8) | 5 (5–7) | 1 (1–3) |
| Anus | 9 (8–9) | 8 (6–8) | 5 (5–7) | 1 (1–3) |
| Head and neck | 9 (7–9) | 7.5 (6–8) | 5 (3–6) | 3 (1–5) |
| Breast | 9 (7–9) | 7.5 (6–8) | 5.5 (4–6) | 1 (1–2) |
| Cervix | 9 (8–9) | 7 (6–8) | 5 (4–6) | 1 (1–3) |
| Melanoma | 9 (8–9) | 7 (6–8) | 4 (3–6) | 1.5 (1–8) |
| Prostate | 8.5 (7–9) | 8 (7–9) | 5.5 (4–8) | 1 (1–5) |
| Sarcoma | 8 (6–8) | 7 (4–7) | 4 (2–5) | 1 (1–2) |
| Ovary | 8 (7–9) | 7 (5–8) | 3 (2–5) | 1 (1–3) |
| Bladder | 8.5 (7–9) | 6.5 (6–8) | 4 (3–5) | 1 (1–2) |
| Urothelial tract | 8 (6–9) | 5.5 (5–7) | 4 (3–5) | 1.5 (1–3) |
| Lung | 7 (6–9) | 5 (3–8) | 3 (1–5) | 1 (1–2) |
| Stomach | 7 (6–8) | 4.5 (2–7) | 2 (1–5) | 1 (1–1) |
| Esophagus | 7 (5–8) | 4 (3–7) | 2 (1–5) | 1 (1–1) |
| Gallbladder | 5 (4–6) | 3 (2–5) | 2 (1–3) | 1 (1–1) |
| Liver/intrahepatic bile duct | 4 (2–7) | 3 (2–7) | 1.5 (1–5) | 1 (1–1) |
| Pancreas | 4 (3–7) | 2.5 (1–5) | 1 (1–3) | 1 (1–1) |

Experts were asked to rate how likely they believed a cancer could be "cured" today in each stage, defined as the receipt of effective treatment such that a population of individuals who are "cured" would have the same life expectancy as a population that never had the cancer being considered. Curability was rated on a scale of 1 to 9: extremely unlikely = 1; somewhat unlikely = 3; neutral, neither likely nor unlikely = 5; somewhat likely = 7; extremely likely = 9.

**Table 2. Estimated median (range) number of years for each cancer type to progress from one stage to the next.**

| Cancer type | Stage I | Stage II | Stage III |
|---|---|---|---|
| **Prostate** | 7 (5–8) | 5 (4–6) | 3 (2–5) |
| **Thyroid** | 5.5 (4–8) | 5 (3–7) | 4 (2–5) |
| **Kidney** | 5 (<1–7) | 3 (<1–5) | 2 (<1–2) |
| **Uterus** | 4 (3–5) | 3 (<1–5) | 1.5 (<1–3) |
| **Cervix** | 4 (<1–5) | 2.5 (<1–4) | <1 (<1–2) |
| **Colon/rectum** | 3.5 (2–5) | 3 (2–5) | <1 (<1–2) |
| **Sarcoma** | 3.5 (<1–6) | 2 (<1–4) | <1 (<1–2) |
| **Breast** | 3 (2–4) | 2 (<1–3) | 1.5 (<1–2) |
| **Melanoma** | 3 (<1–5) | 2 (<1–4) | <1 (<1–2) |
| **Head and neck** | 3 (2–6) | 2 (<1–4) | <1 (<1–2) |
| **Bladder** | 3 (2–5) | 2 (<1–5) | <1 (<1–2) |
| **Ovary** | 3 (<1–3) | 2 (<1–2) | <1 (<1-<1) |
| **Stomach** | 3 (2–5) | 2 (<1–2) | <1 (<1–2) |
| **Urothelial tract** | 3 (2–7) | 2 (2–5) | <1 (<1–4) |
| **Anus** | 3 (2–7) | 2 (2–5) | <1 (<1–3) |
| **Esophagus** | 2.5 (2–5) | <1 (<1–2) | <1 (<1–2) |
| **Lung** | 2 (2–3) | <1 (<1–2) | <1 (<1-<1) |
| **Liver/intrahepatic bile duct** | 2 (<1–3) | <1 (<1–2) | <1 (<1-<1) |
| **Gallbladder** | 2 (<1–3) | <1 (<1-<1) | <1 (<1-<1) |
| **Pancreas** | <1 (<1–2) | <1 (<1–2) | <1 (<1-<1) |

Experts were asked to rate how long cancers would take to progress from the beginning of one stage to the beginning of the next. Progression was rated on a scale of 1 (less than a year) to 9 (9 or more years).

in stage III, 0% in stage IV. More stage I cancers were rated as being somewhat likely to extremely likely to be cured than later-stage cancers.

Expert estimates of preclinical cancer progression are provided in Table 2. Prostate and thyroid cancer were estimated to be the slowest growing, taking approximately 7 and 5 years, respectively, to progress through stage I (range, 4–8), 5 years to progress through stage II (range, 3–7), and 3 and 4 (range, 2–5) years, respectively, to progress through stage III. Esophageal, lung, liver/intrahepatic bile duct, gallbladder, and pancreatic cancers were estimated to progress quickly through stages I to III (1 to 2 years per stage). The estimated likelihood of cancer curability today in earlier stages and likelihood of progression to the next stage are depicted in S1 Fig.

## Benefit from early detection

When experts considered whether cancers would benefit from early detection by a hypothetical screening test that could detect cancers at all stages, 3 groups of cancers emerged (illustrated in the different colors in Fig 1). Experts estimated those cancers that progress quickly and are considered currently curable would benefit the most from early detection. These cancer types (75.0% of all cancers rated) included stomach, esophagus, lung, urothelial tract, melanoma, ovary, sarcoma, bladder, cervix, breast, colon/rectum, kidney, uterus, anus, and head and neck. Cancer types that progress but were considered to be less curable (liver/intrahepatic bile duct, gallbladder, pancreas) were rated as potentially showing some benefit. Finally, experts did not expect cancers that progress slowly and are curable (prostate, thyroid) to benefit from early detection.

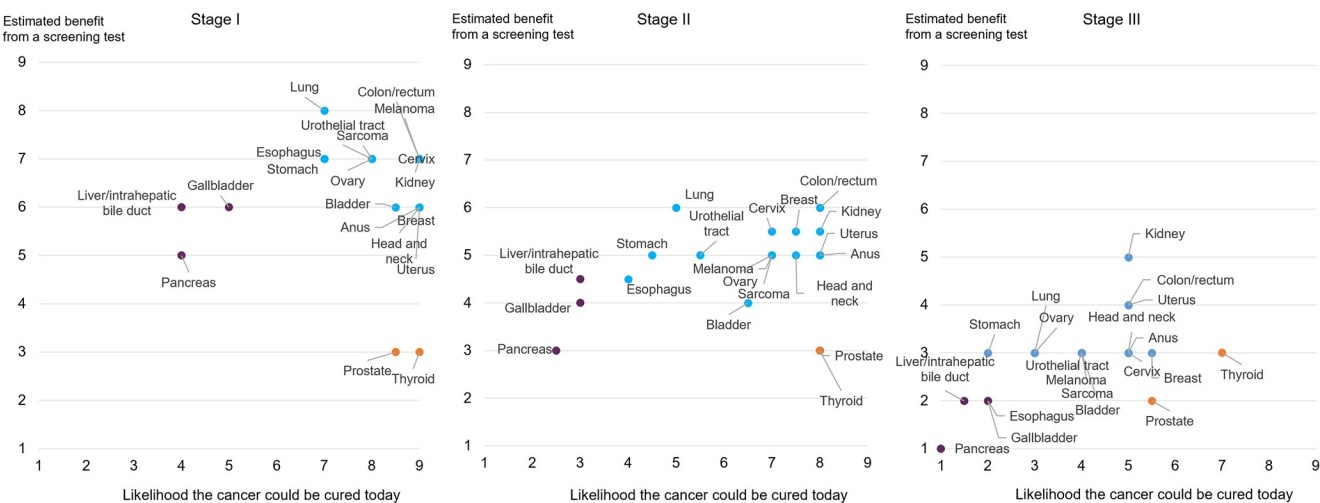

**Fig 1. Ratings of cancer curability today and the estimated benefit from a hypothetical screening blood test.** The x-axis represents expert ratings on how likely they believe a cancer could be "cured" today, defined as the receipt of effective treatment such that a population of individuals who are "cured" would have the same life expectancy as a population that never had the cancer being considered. Curability was rated on a scale of 1 (extremely unlikely to be cured) to 9 (extremely likely to be cured). The y-axis represents expert ratings on the estimated benefit from a hypothetical screening test, defined as the improvement in curability as a result of the test. Improvement in cure was rated on a scale of 1 (not at all likely to increase) to 9 (increase a great deal). Cancers in blue are those experts believed could potentially benefit the most from a hypothetical screening test (those with high curability today in earlier stages and likely to progress). Experts believed cancers in purple may show some benefit from a hypothetical screening test (lower curability today in earlier stages and likely to progress). Cancers in orange were rated as least likely to benefit from a hypothetical screening test (high curability in earlier stages today and progress slowly).

## Typical treatment versus best available care

Panelists generally rated a higher benefit from early detection if best-available versus typical care was provided (Fig 2, see liver/intrahepatic bile duct, gallbladder, and pancreatic cancers). The difference in potential benefit of early detection between typical versus best-available care was generally higher in stages II and III.

## Discussion

The panel concluded that most solid tumors would have better treatment outcomes if detected early. Only prostate and thyroid cancer, both of which have good long-term survival when diagnosed early, were not expected to benefit from earlier detection by a hypothetical blood-based screening test.

Panelists agreed that the cancers most likely to benefit from early detection were those that progress quickly and are currently curable in earlier stages: anus, bladder, breast, cervix, colon/rectum, esophagus, head and neck, kidney, lung, melanoma, ovary, sarcoma, stomach, urothelial tract, and uterus. Many of these cancers do not have established screening tests, and curability varies by stage at diagnosis. For example, in Fig 1, stage I ovarian cancer was rated as having a high likelihood of curability, whereas in stage III, the likelihood of curability was very low. A similar pattern can be seen in the curability ratings for sarcoma and bladder cancer, both of which do not have established screening tests. The ability to screen for multiple cancers at once, particularly for cancers without existing screening tests, could identify cancer in patients before it is clinically evident. Expert opinion therefore concurs with the intuition that detection at an earlier stage of cancer could improve patient outcomes by diagnosing cancer when it is potentially more treatable.

Experts agreed that pancreatic, gallbladder, and liver/intrahepatic bile duct cancer would benefit from a hypothetical screening blood test, though to a lesser degree than the cancers

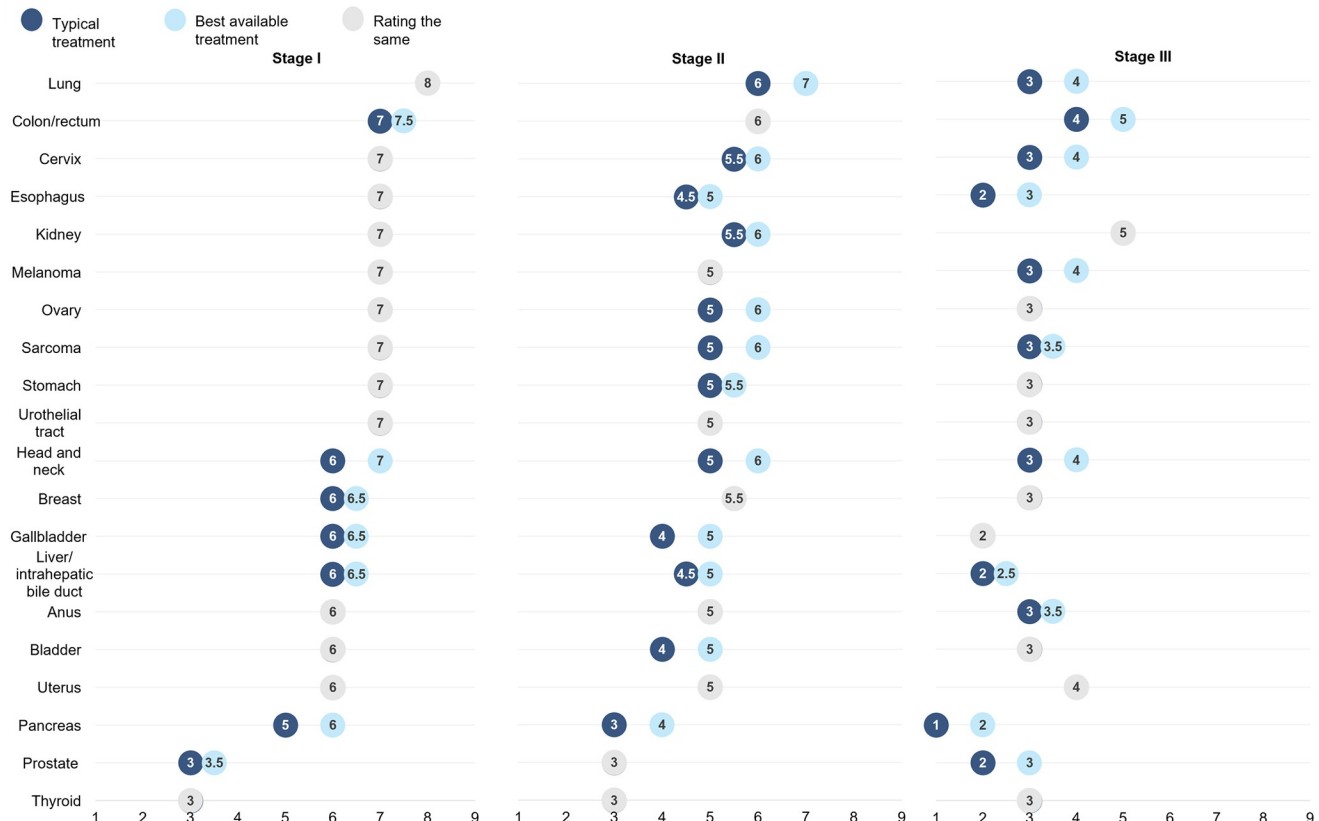

**Fig 2. Estimated benefit from a hypothetical screening blood test with typical versus best-available treatment.** This figure illustrates expert ratings on the estimated benefit from a hypothetical screening test, defined as the improvement in curability as a result of the test on a scale of 1 (not at all likely to increase) to 9 (increase a great deal), when considering typical versus best available treatment. Typical treatment was defined as the care provided to the population as a whole; best available treatment was defined as guideline-concordant care. Items in grey represent no difference in ratings for typical versus best available treatment. Items in blue represent a difference in ratings for typical versus best available treatment, with the dark blue representing typical treatment and light blue representing best available treatment.

listed above. These cancers do not have established screening tests and currently have unfavorable survival statistics. Based on SEER18 registry data, the median 10-year survival for stage II pancreatic, gallbladder, and liver/intrahepatic bile duct cancer is 10%, 19%, and 30%, respectively. If diagnosed in stage I, the 10-year survival is higher (pancreatic cancer: 31%; gallbladder: 47%; liver/intrahepatic bile duct: 32%), likely due to more effective treatments [21–23]. Currently, the majority of patients with these cancers are diagnosed at later stages, which may be limiting the opportunities to develop more effective early-stage treatments. By reducing the prevalence of tumors diagnosed at later stages, potential research gains along with significant improvements in survival could be made.

Prostate and thyroid cancer were not considered good candidates for a hypothetical screening blood test. These cancers have good long-term survival, especially in stages I and II, and even in stage III, though to a lesser extent. These findings align with current practice. The USPSTF recommends against screening the general asymptomatic adult population for thyroid cancer as the harms of screening were considered to outweigh any potential benefits [24]. While the incidence of thyroid cancer steadily increased, mortality rates have remained stable since 1975, suggesting screening may be primarily identifying cancers that would not result in death [25]. The USPSTF also recommends against screening for prostate cancer in men older

than 70 years and in those aged 55 to 69 years who do not express a preference for screening [26]. While PSA screening has been shown to reduce metastatic prostate cancer and disease-related death, there are downsides to screening including false-positive tests leading to unnecessary biopsies, overdetection of insignificant cancers, and treatment-related complications [27, 28]. Several organizations (i.e., American Cancer Society, American Urological Association, American College of Physicians) have recommended shared decision making about PSA screening. Risk-adapted individualized approaches including using PSA in conjunction with risk factors (e.g., genetics, race, family history), as well as incorporation of advanced magnetic resonance imaging (MRI) may improve the detection of significant prostate cancer and avoid the overdiagnosis of clinically insignificant disease [29].

Detecting cancers at earlier stages can be crucial to increasing survival rates because it increases the likelihood of treating cancer when more treatment options are available. There have been significant improvements in available treatment for early-stage breast, cervical, and lung cancer over the last few decades, translating screening for these cancers into decreased mortality [30–32]. Experts expected treatment innovations would continue, especially in liver/intrahepatic bile duct, gallbladder, and pancreatic cancers, which they anticipated could lead to improved outcomes if cancers were identified through screening. In addition, experts estimated the time to progression from one stage to the next to be faster with each subsequent stage (e.g., 3 years from stage I to stage II, 1 year from stage II to stage III), suggesting there could be more opportunity to detect and intervene at earlier stages and highlighting the need for a screening blood test to be sensitive enough to detect cancers early.

Cancer treatment is not always optimal. Guideline-concordant care was estimated to be provided in approximately 80% of prostate cancers [33, 34], 75% to 80% of breast and endometrial cancers [18, 35–37], 62% of cervical cancers [20], and 44% of lung cancers [38]. Experts felt that most cancers, especially stages II and III, would benefit more from early detection if also treated with best-available multidisciplinary care, including those that were not expected to benefit from a hypothetical screening test with typical care (e.g., prostate).

Clinicians must weigh the benefits and harms of screening. Even minor reductions in screening specificity can be associated with large increases in false-positive rates, costs, and other associated harms for low-prevalence cancers [39]. While multi-cancer screening blood tests may also result in false-positive findings [3] and a fixed base of harms, these harms may be substantially reduced compared to the harms associated with screening one cancer at a time [40]. Additionally, the convenience and accessibility of a single multi-cancer screening blood test could improve adherence to existing screening, which is often suboptimal [41–43].

This study had several limitations. First, these results reflect the opinion of 10 experts, one of whom was a general oncologist, though the remaining experts represented the majority of cancer types covered within this study. However, we gathered these opinions using the RAND/UCLA modified Delphi panel method, a validated quantitative method for eliciting expert opinion, which has been used extensively to develop quality measures and clinical guidance in a variety of areas [44]. Ratings of appropriateness from this method have been found to be reliable with test-retest reliability >0.9 using the same panelists 6 to 8 months later [45] and kappa statistics across several panels with different members similar to those of some common diagnostic tests [46]. Second, we used expert opinion to identify cancers that may benefit from early detection, rather than collecting data objectively on whether mortality rates changed from screening tests. Finally, the experts included in the panel were from the US only. Therefore, their opinions were based on their knowledge of cancer incidence and mortality rates within the US, which may vastly differ from the opinions of experts from undeveloped or developing countries, given the cancer burden may be distinct in areas with substantial variation in socioeconomic and cultural characteristics.

We also asked experts to assume the hypothetical screening test had 100% sensitivity and 100% specificity to simplify the rating process (e.g., base ratings on clinical factors without accounting for test performance) and focus attention on the effects of early detection rather than the methods, despite differences in the real-world performance of a screening test [47]. Further, although panelists discussed that morbidity is important and could be impacted by screening tests, only the impact of screening tests on mortality was evaluated in the rating form. Lastly, certain cancer groupings used encapsulated a number of disparate cancer types and varying risk profiles, making their ratings of these cancer types challenging. We recommend including further granularity of cancer subtypes in future research.

## Conclusions

The panel highlighted opportunities to improve cancer cure by intercepting cancers at an earlier stage. Both clinical trials and real-world evidence have demonstrated that early detection is associated with better survival. Even among difficult-to-treat cancers (e.g., pancreas, liver/intrahepatic bile duct, gallbladder), early-stage detection was believed to be beneficial. Based on the panel consensus, increased coverage of cancer types in a screening test would deliver the greatest potential benefits to patients.

## Supporting information

**S1 Fig. Estimated cancer curability at a given stage and progression to the next stage.**
(PDF)

**S1 Table. Pre-meeting (round 1) aggregated ratings from all panelists.**
(PDF)

**S2 Table. Post-meeting (round 2) aggregated ratings from all panelists.**
(PDF)

## Author Contributions

**Conceptualization:** Michael S. Broder, Anuraag Kansal, Ashley Kim, Irina Yermilov.

**Data curation:** Michael S. Broder, Cynthia Campos, Irina Yermilov.

**Formal analysis:** Michael S. Broder, Cynthia Campos, Irina Yermilov.

**Methodology:** Michael S. Broder, Anuraag Kansal, Ashley Kim, Irina Yermilov.

**Validation:** Michael S. Broder, Cynthia Campos, Irina Yermilov.

**Visualization:** Michael S. Broder, Cynthia Campos, Irina Yermilov.

**Writing – original draft:** Michael S. Broder, Anuraag Kansal, Ashley Kim, Irina Yermilov.

**Writing – review & editing:** Lee Schwartzberg, Sikander Ailawadhi, Himisha Beltran, L. Johnetta Blakely, G. Thomas Budd, Laurie Carr, Michael Cecchini, Patrick Cobb, Anuraag Kansal, Ashley Kim, Bradley J. Monk, Deborah J. Wong, Cynthia Campos, Irina Yermilov.

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
