## [Decision Letter · Decision Letter 0]

19 Jul 2022

PONE-D-22-14922Impact of early detection on cancer curability: a modified Delphi panel studyPLOS ONE

Dear Dr. Kim,

Thank you for submitting your manuscript to PLOS ONE. After careful consideration, we feel that it has merit but does not fully meet PLOS ONE’s publication criteria as it currently stands. Therefore, we invite you to submit a revised version of the manuscript that addresses the points raised during the review process.

We look forward to receiving your revised manuscript.

Kind regards,

Kush Raj Lohani, Master of Surgery

Academic Editor

PLOS ONE

Journal Requirements:

"I have read the journal's policy and the authors of this manuscript have the following competing interests: LS, SA, HB, JB, GTB, LC, MC, PC, BJM, DJW reported consulting fees from Partnership for Health Analytic Research (PHAR), LLC and GRAIL, LLC, a subsidiary of Illumina, Inc., currently held separate from Illumina, Inc. under the terms of the Interim Measures Order of the European Commission dated 29 October 2021. ARK and AK are employees of GRAIL, LLC, a subsidiary of Illumina, Inc., currently held separate from Illumina, Inc. under the terms of the Interim Measures Order of the European Commission dated 29 October 2021, who supported this study, and reported stock or stock options in Illumina (parent company of GRAIL, LLC, a subsidiary of Illumina, Inc., currently held separate from Illumina, Inc. under the terms of the Interim Measures Order of the European Commission dated 29 October 2021). MSB, IY, and CC are employees of Partnership for Health Analytic Research (PHAR), LLC which was paid by the following to conduct research related to the work described in the manuscript: Abbvie, Amgen, AstraZeneca, Biomarin Pharmaceuticals, BMS, Celgene, Dompe, Eisai, Exact Sciences Corporation, Genentech, GRAIL, LLC, a subsidiary of Illumina, Inc., currently held separate from Illumina, Inc. under the terms of the Interim Measures Order of the European Commission dated 29 October 2021, Ionis, Jazz, Kite, Novartis, Otsuka, Recordati, Regeneron, Sanofi US Services, Takeda Pharmaceuticals USA."

Reviewers' comments:

Reviewer's Responses to Questions

**Comments to the Author**

1. Is the manuscript technically sound, and do the data support the conclusions?

Reviewer #1: Partly

Reviewer #2: Yes

2. Has the statistical analysis been performed appropriately and rigorously? 

Reviewer #1: No

Reviewer #2: N/A

3. Have the authors made all data underlying the findings in their manuscript fully available?

Reviewer #1: No

Reviewer #2: No

4. Is the manuscript presented in an intelligible fashion and written in standard English?

Reviewer #1: Yes

Reviewer #2: Yes

5. Review Comments to the Author

Reviewer #1: This is an interesting paper on a modified Delphi study on the impact of early detection on cancer mortality. However, some parts of the methodology are not fully clear / justified in the current version, specifically:

1) Data access: only aggregated data is provided in the manuscript, without individual expert scores; and only the results of the second round of scoring are reported. It is suggested to include the individual expert ratings for both rating rounds as supplementary material.

2) Lines 105-107: The Delphi panel involved 10 oncology experts from a diversity of oncology subspecialties (e.g., prostate, breast, lung, colorectal, gastrointestinal, gynecologic, head and neck, hematologic cancers, and sarcoma) and included a general practice oncologist. Apparently, a single expert was involved from most subspecialties, raising that the consensus building on the specific cancer types could be dominated by this single expert with the strongest background in the corresponding subspecialty. The Authors are invited to discuss this risk/limitation, and to report more details on disagreements in round 1 that were solved in round 2.

3) Word anchors for benefit were 1 (cure rates not at all likely to increase) and 9 (cure rates likely to increase a great deal). Please clarify if additional word anchors were used for scores 2-8.

4) Experts were asked to rate curability of 20 solid organ cancers across stages I to IV before the meeting, where they were presented with individual first-round ratings and median values, and the reasons of their peers for their ratings (Lines 143-154). On the other hand, SEER registry data on median 10-year survival was also referenced in the manuscript by cancer types and stages (Lines 256-260). The Authors are invited to explain why did they decide to collect curability data from their expert panel of limited size and heterogeneous subspecialty background, instead of building on SEER registry data; and whether SEER data on curability / survival by cancer types and stages was presented at the consensus meeting to the involved panelists or not. Correlation of expert-elicited curability rates with SEER statistics on survival would also be an interesting point to add to this manuscript as a kind of validation, even if curability (as elicited by the experts) and 10-year survival (in SEER analyses) are not identical concepts.

5) Table 2, estimated median (range) number of years for each cancer type to progress from one stage to next: The Authors are invited to clarify whether there is existing data in the literature / in registries relevant for the US on cancer stage transition times, or not. The lack of available data would justify the use of the opinions of the 10 involved experts as data source.

6) Figure 1 shows three clusters of cancer types-stages, by various colors: cancers in blue ("those with high curability today in earlier stages and likely to progress"); cancers in purple ("lower curability today in earlier stages

and likely to progress"); and cancers in orange ("high curability in earlier stages today and progress slowly"). However, it is unclear how these clusters were formed. If statistical methods were used for the identification of clusters, the corresponding methods should be described. If the clusters were identified by the Authors visually, please provide a corresponding chart that allows for visual cluster identification (i.e., a plot of curability today ~ progression to next stage), since Figure 1 does not include data on likeliness to progress. There is a risk that the clusters were defined based on preconception - which is a very plausible preconception, but is not driven from the expert panel score data.

Please note that figure resolution need to be improved.

Reviewer #2: Disclosures: ok

Title: adequate

Key words: There is not available key words, please insert it

Abstract: adequate

Introduction

PG 3- line 63- Please inset some words about PSA, however controverse there are in the literature several prostate cancer, screening studies using PSA as a serum marker. I suggest to include it, with imaging tests, tissue tests and biopsy. Despite to promote overdiagnosis and overtreatment, and the elevated number necessary to treat patients, European Trials has shown death reduction in prostate cancer screened patients.

I think prostate cancer screening controversies might be cited in paragraphs between lines 77-91.

Methods:

Please, clarify how were the criteria for the 10 experts invitation. It is not clear in the methodology (or how many were invited and declined e.g.), How to choose someone in a US region, among a great quantity of oncologists available?

Results

Figures present poor quality; being necessary to click in the link ( good quality at link). It is necessary to enhance it for the readers.

Figures legend are absent it he manuscript- Are there in the pages 11 and 12?

Discussion

In the study limitations, authors must recognize that they focused only in the United States scenario. We do not know what would be the opinion of experts form Far East Countries, or form undeveloped and developing countries. As we know the incidence and mortality rates from several prevalent tumors, are quite distinct among several nations and in distinct socioeconomical and cultural populational groups.

It is difficult to understand why only 10 experts were choose for this study. In my view this study might be a pilot or a feasibility study, performed as a first step, to development of a new screening, which could invite hundreds of experts around the world, and to include radiation oncologists, oncologic surgeons etc. What was the objective of authors in inviting on 10 professionals, in a where the answers may be contaminated by subjectively and diverse personal bias?

The hypothetic blood test only diagnostic early stage cancers or cancer in all stages? Please clarify it.

6. PLOS authors have the option to publish the peer review history of their article (what does this mean?). If published, this will include your full peer review and any attached files.

Reviewer #1: **Yes: **Janos G Pitter MD, PhD

Reviewer #2: No

---

## [Author Response · Author response to Decision Letter 0]

20 Sep 2022

Thank you for your in-depth review and helpful comments. Based on the feedback provided, we have made edits to the manuscript and have included our responses below to the reviewers’ comments. The new changes have been identified in the revised manuscript.

Reviewer's Responses to Questions

Comments to the Author

1. Is the manuscript technically sound, and do the data support the conclusions?

Reviewer #1: Partly

Reviewer #2: Yes

2. Has the statistical analysis been performed appropriately and rigorously?

Reviewer #1: No

Reviewer #2: N/A

3. Have the authors made all data underlying the findings in their manuscript fully available?

Reviewer #1: No

Reviewer #2: No

4. Is the manuscript presented in an intelligible fashion and written in standard English?

Reviewer #1: Yes

Reviewer #2: Yes

5. Review Comments to the Author

Reviewer #1: This is an interesting paper on a modified Delphi study on the impact of early detection on cancer mortality. However, some parts of the methodology are not fully clear / justified in the current version, specifically:

1) Data access: only aggregated data is provided in the manuscript, without individual expert scores; and only the results of the second round of scoring are reported. It is suggested to include the individual expert ratings for both rating rounds as supplementary material.

Response: Thank you for this comment. We have added the aggregated raw results from both the first and second rounds as supplemental material. Experts were told their individuals scores would not be shared with others, and only group medians were shared with the panel during the discussion. Given our panel of experts was small, there may be concerns regarding the privacy of the panelists if we share anonymized individual scores. We have noted these points in the cover letter.

2) Lines 105-107: The Delphi panel involved 10 oncology experts from a diversity of oncology subspecialties (e.g., prostate, breast, lung, colorectal, gastrointestinal, gynecologic, head and neck, hematologic cancers, and sarcoma) and included a general practice oncologist. Apparently, a single expert was involved from most subspecialties, raising that the consensus building on the specific cancer types could be dominated by this single expert with the strongest background in the corresponding subspecialty. The Authors are invited to discuss this risk/limitation, and to report more details on disagreements in round 1 that were solved in round 2.

Response: Thank you for this point. The panel consisted of ten oncology experts representing various clinical specialties and a diverse range of experience and backgrounds. Given the level of expertise of most, if not all, of the specialists, we rarely had a single expert speaking on a specific cancer type, as there was a high level of familiarity with regards to multiple cancer types as a whole. Panelists were explicit on whether they had any experience with a certain cancer type, and the discussion stemmed naturally from the panelists comparing their experiences with different cancer types and patient populations more broadly, as well as different treatment options and severity levels. Most of the disagreements in round 1 that were resolved in round 2 stemmed from panelists having various interpretations of some of the questions and concepts on the rating form, as they discussed how to define and consider “curability” and cure rates, life expectancy, and staging of cancers. 

3) Word anchors for benefit were 1 (cure rates not at all likely to increase) and 9 (cure rates likely to increase a great deal). Please clarify if additional word anchors were used for scores 2-8.

Response: Thank you for this comment. To clarify, when asking panelists whether cure rates would increase with a hypothetical screening test, we provided a scale of 1-9, with 1 (cure rates are not at all likely to increase), 3 (cure rates are slightly likely to increase), 5 (cure rates are moderately likely to increase), 7 (cure rates are considerably likely to increase), and 9 (cure rates are likely to increase a great deal, or double). We did not provide more specific and tangible word anchors for the other scores other than a score of 9, as we hypothesized that it would be difficult for panelists to think about a certain percentage (e.g., 10-20%) of the outcome of interest for each cancer type and stage. Instead, we used categories which allowed more room for panelists to answer the question in a more holistic way, given the nuances with each cancer type and stage and the highest possible score of 9 would suggest doubling. 

4) Experts were asked to rate curability of 20 solid organ cancers across stages I to IV before the meeting, where they were presented with individual first-round ratings and median values, and the reasons of their peers for their ratings (Lines 143-154). On the other hand, SEER registry data on median 10-year survival was also referenced in the manuscript by cancer types and stages (Lines 256-260). The Authors are invited to explain why did they decide to collect curability data from their expert panel of limited size and heterogeneous subspecialty background, instead of building on SEER registry data; and whether SEER data on curability / survival by cancer types and stages was presented at the consensus meeting to the involved panelists or not. Correlation of expert-elicited curability rates with SEER statistics on survival would also be an interesting point to add to this manuscript as a kind of validation, even if curability (as elicited by the experts) and 10-year survival (in SEER analyses) are not identical concepts.

Response: Thank you for this comment. Prior to the consensus meeting, the panelists were provided a comprehensive evidence summary on the survivability (curability) of 20 solid organ cancers, which was primarily obtained from SEER data and the American Cancer Society (ACS) Facts & Figures 2020 Report. We wanted to elicit expert input on curability across cancer stages and types based on their current clinical practice and experience, and also on curability given the addition of a hypothetical screening test and either typical or best available treatment. 

5) Table 2, estimated median (range) number of years for each cancer type to progress from one stage to next: The Authors are invited to clarify whether there is existing data in the literature / in registries relevant for the US on cancer stage transition times, or not. The lack of available data would justify the use of the opinions of the 10 involved experts as data source.

Response: Thank you for this point. There is limited data and literature to allow inferences regarding the natural history of cancers (e.g., 5-year survival by stage, tumor doubling time) for a few cancer types, but there is a lack of relevant data for the large majority of cancer types. Even for cancer types with some data on natural history, there is an absence of data on progression between stages prior to clinical diagnosis.

6) Figure 1 shows three clusters of cancer types-stages, by various colors: cancers in blue ("those with high curability today in earlier stages and likely to progress"); cancers in purple ("lower curability today in earlier stages

and likely to progress"); and cancers in orange ("high curability in earlier stages today and progress slowly"). However, it is unclear how these clusters were formed. If statistical methods were used for the identification of clusters, the corresponding methods should be described. If the clusters were identified by the Authors visually, please provide a corresponding chart that allows for visual cluster identification (i.e., a plot of curability today ~ progression to next stage), since Figure 1 does not include data on likeliness to progress. There is a risk that the clusters were defined based on preconception - which is a very plausible preconception, but is not driven from the expert panel score data.

Response: Thank you for this comment. The three clusters of cancer types were in fact determined from the panel consensus on the estimated likelihood of cure by cancer type and stage (Table 1) and the estimated number of years for cancer to progress through stages (Table 2), both of which are in the manuscript. Panelists agreed that cancers in orange (e.g., prostate, thyroid) progress slowly despite having high curability in earlier stages. Panelists also agreed that cancers in purple (e.g., gallbladder, live/intrahepatic bile duct, and pancreas) are likely to progress, but have a lower likelihood of cure today in earlier stages. Finally, panelists agreed that cancers in blue (e.g., anus, uterus, head and neck, kidney, colon/rectum, breast, cervix, urothelial tract, lung, melanoma, ovary, sarcoma, bladder, stomach, esophagus) are likely to progress, but have a lower likelihood of cure today in earlier stages. Also, we have added Figure S1, which plots the likelihood of cancer curability today and the progression to the next stage, as part of the Supporting Information and in lines 185-186. 

Please note that figure resolution need to be improved.

Response: Thank you for this comment. We have reuploaded the figures to PACE.

Reviewer #2: Disclosures: ok

Title: adequate

Key words: There is not available key words, please insert it

Response: Thank you for this comment. The following key words (cancer screening, Delphi panel, survival, curability) have been added within the submission form. 

Abstract: adequate

Introduction

PG 3- line 63- Please inset some words about PSA, however controverse there are in the literature several prostate cancer, screening studies using PSA as a serum marker. I suggest to include it, with imaging tests, tissue tests and biopsy. Despite to promote overdiagnosis and overtreatment, and the elevated number necessary to treat patients, European Trials has shown death reduction in prostate cancer screened patients.

I think prostate cancer screening controversies might be cited in paragraphs between lines 77-91.

Response: Thank you for this comment. We included PSA tests as an example of an existing cancer screening test, albeit on an individual basis, in lines 64-65. We did not include other imaging tests, tissue tests, and biopsies, as the scope of this study was to consider cancer screening modalities specifically, and not diagnostic tests. We have also elaborated further on the controversies surrounding PSA screening in lines 281-293.

Methods:

Please, clarify how were the criteria for the 10 experts invitation. It is not clear in the methodology (

or how many were invited and declined e.g.), How to choose someone in a US region, among a great quantity of oncologists available?

Response: Thank you for this comment. Given readily accessible lists of experts across the US were unavailable, recruitment was conducted indirectly (e.g., professional networks). We initially reached out to 37 experts across the US to gauge interest and availability. We ended up with a panel of 11 experts, of which one was not able to attend the meeting. As a result, our final panel included 10 experts, which was in line with the RAND/UCLA Appropriateness Method guideline (Fitch et al 2001) recommendations of 7-15 panelists. The criteria for the 10 experts included having a breadth and diversity of oncology experience (i.e., treating a variety of cancers) across the US, representing different geographic regions and practice settings. This has been clarified in lines 102-107. 

Results

Figures present poor quality; being necessary to click in the link ( good quality at link). It is necessary to enhance it for the readers.

Response: Thank you for this comment. The figures have been reuploaded to PACE. 

Figures legend are absent it he manuscript- Are there in the pages 11 and 12?

Response: Thank you for pointing this out. The Figure 1 legend is within lines 210-223, and the Figure 2 legend is within lines 233-241.

Discussion

In the study limitations, authors must recognize that they focused only in the United States scenario. We do not know what would be the opinion of experts form Far East Countries, or form undeveloped and developing countries. As we know the incidence and mortality rates from several prevalent tumors, are quite distinct among several nations and in distinct socioeconomical and cultural populational groups.

Response: Thank you for this point. We recognize our US-specific a panel as a study limitation in lines 330-335. 

It is difficult to understand why only 10 experts were choose for this study. In my view this study might be a pilot or a feasibility study, performed as a first step, to development of a new screening, which could invite hundreds of experts around the world, and to include radiation oncologists, oncologic surgeons etc. What was the objective of authors in inviting on 10 professionals, in a where the answers may be contaminated by subjectively and diverse personal bias?

Response: Thank you for your comment. We chose 10 panelists because this is in line with the RAND/UCLA Appropriateness Method guidelines (Fitch et al 2001), which recommended anywhere from 7-15 panelists. We initially reached out to 37 experts to gauge interest and availability. The final 10 experts for the panel were chosen due to their breadth and diversity of oncology experience across the US, with the aim of including oncologists who treat a variety of cancers. This has been clarified in lines 102-107. In addition, given the level of expertise of most, if not all, of the specialists, we rarely had a single expert speaking on a specific cancer type, as there was a decent level of overlap and familiarity with regards to multiple cancer types as a whole. Panelists were explicit on whether they did or did not have much experience with a certain cancer type, and the discussion stemmed naturally from the panelists comparing their experiences with different cancer types and patient populations more broadly, as well as different treatment options and severity levels. Most of the disagreements stemmed from panelists having various interpretations of some of the questions and concepts on the rating form, as they discussed how to define and consider “curability” and cure rates, life expectancy, and staging of cancers.

The hypothetic blood test only diagnostic early stage cancers or cancer in all stages? Please clarify it.

Response: Thank you for this question. Our hypothetical scenario of annual cancer screening blood test with high sensitivity and specificity would be able to detect cancers at all stages. We have clarified this point in line 197.

6. PLOS authors have the option to publish the peer review history of their article (what does this mean?). If published, this will include your full peer review and any attached files.

Do you want your identity to be public for this peer review? For information about this choice, including consent withdrawal, please see our Privacy Policy.

Reviewer #1: Yes: Janos G Pitter MD, PhD

Reviewer #2: No

Response: The figure files have been uploaded to PACE.

---

## [Decision Letter · Decision Letter 1]

6 Nov 2022

PONE-D-22-14922R1Impact of early detection on cancer curability: a modified Delphi panel studyPLOS ONE

Dear Dr. Authors,

Thank you for submitting your manuscript to PLOS ONE. After careful consideration, we feel that it has merit but does not fully meet PLOS ONE’s publication criteria as it currently stands. Therefore, we invite you to submit a revised version of the manuscript that addresses the points raised during the review process.

We look forward to receiving your revised manuscript.

Kind regards,

Kush Raj Lohani, MBBS, MS (Master of General Surgery)

Academic Editor

PLOS ONE

Journal Requirements:

Additional Editor Comments (if provided):

Thank you for this interesting and conceptual study. There is no doubt that early diagnosis and prompt treatment remains our main motto. However, before we jump into universal screening for all cancers, we need to know the limitations and practical aspects of it. We certainly need to weigh benefit / risk of screening to provide comprehensive benefit to patients. This study is an initial step in this endeavor. As highlighted in the reviews, one of the main limitations of this study is incorporating opinion of an Oncologist on all cancers. I hope this will be mentioned in the limitations. I believe this study will invite further dedicated studies with more concrete data from experts in each type of cancer by implementing real data from the institutional experience and comparing with reference information such as from the SEER.

Reviewers' comments:

Reviewer's Responses to Questions

**Comments to the Author**

1. If the authors have adequately addressed your comments raised in a previous round of review and you feel that this manuscript is now acceptable for publication, you may indicate that here to bypass the “Comments to the Author” section, enter your conflict of interest statement in the “Confidential to Editor” section, and submit your "Accept" recommendation.

Reviewer #1: (No Response)

2. Is the manuscript technically sound, and do the data support the conclusions?

Reviewer #1: Yes

3. Has the statistical analysis been performed appropriately and rigorously? 

Reviewer #1: Yes

4. Have the authors made all data underlying the findings in their manuscript fully available?

Reviewer #1: Yes

5. Is the manuscript presented in an intelligible fashion and written in standard English?

Reviewer #1: Yes

6. Review Comments to the Author

Reviewer #1: The Authors explained in their response letter that panelists were explicit on whether they had any

experience with a certain cancer type or not. This information is important for the interpretation of the findings and shall be added to the manuscript: please include the number of experts with explicit experience for each specific cancer type.

In addition, please clarify whether scores were collected from all experts for all cancer types? Or for each cancer type, scores were collected only from experts with declared experience with the particular cancer type?

7. PLOS authors have the option to publish the peer review history of their article (what does this mean?). If published, this will include your full peer review and any attached files.

Reviewer #1: **Yes: **Janos G. Pitter MD, PhD

---

## [Author Response · Author response to Decision Letter 1]

21 Nov 2022

Thank you for your review and comments. Based on the feedback provided, we have made edits to the manuscript and have included our responses below to the reviewers’ comments. The new changes have been identified in the revised manuscript.

Journal Requirements:

Response: No changes need to be made to the reference list. 

Additional Editor Comments (if provided):

Thank you for this interesting and conceptual study. There is no doubt that early diagnosis and prompt treatment remains our main motto. However, before we jump into universal screening for all cancers, we need to know the limitations and practical aspects of it. We certainly need to weigh benefit / risk of screening to provide comprehensive benefit to patients. This study is an initial step in this endeavor. As highlighted in the reviews, one of the main limitations of this study is incorporating opinion of an Oncologist on all cancers. I hope this will be mentioned in the limitations. I believe this study will invite further dedicated studies with more concrete data from experts in each type of cancer by implementing real data from the institutional experience and comparing with reference information such as from the SEER.

Response: Thank you for your comments. We have included as a limitation that the panel of experts did include one general oncologist, though we have also further clarified that the rest of the panel did represent the majority of cancer types covered within this study in lines 322-324 of the discussion section.

Reviewers' comments:

Reviewer's Responses to Questions

Comments to the Author

1. If the authors have adequately addressed your comments raised in a previous round of review and you feel that this manuscript is now acceptable for publication, you may indicate that here to bypass the “Comments to the Author” section, enter your conflict of interest statement in the “Confidential to Editor” section, and submit your "Accept" recommendation.

Reviewer #1: (No Response)

2. Is the manuscript technically sound, and do the data support the conclusions?

Reviewer #1: Yes

3. Has the statistical analysis been performed appropriately and rigorously?

Reviewer #1: Yes

4. Have the authors made all data underlying the findings in their manuscript fully available?

Reviewer #1: Yes

5. Is the manuscript presented in an intelligible fashion and written in standard English?

Reviewer #1: Yes

6. Review Comments to the Author

Reviewer #1: The Authors explained in their response letter that panelists were explicit on whether they had any

experience with a certain cancer type or not. This information is important for the interpretation of the findings and shall be added to the manuscript: please include the number of experts with explicit experience for each specific cancer type.

In addition, please clarify whether scores were collected from all experts for all cancer types? Or for each cancer type, scores were collected only from experts with declared experience with the particular cancer type?

Response: Thank you for these points. While we had one general practice oncologist, the remaining specialists specified their expertise within the following cancer types: hematologic (2), prostate (1), breast (3), lung (2), colorectal (2), gastrointestinal, including, but not limited to, bile duct, pancreas, esophageal, stomach (1), head and neck (1), liver (1), gynecological, including, but not limited to, cervical and ovarian (1); sarcoma (2). The scores were collected for all cancer types from all experts. As expected in the Delphi approach, the discussion did include the group hearing from experts with more experience for certain cancer types and then each panelist synthesizing that information with their own experience in their individual ratings. We addressed these two points within the methods section in lines 112-115 and 136.

7. PLOS authors have the option to publish the peer review history of their article (what does this mean?). If published, this will include your full peer review and any attached files.

Do you want your identity to be public for this peer review? For information about this choice, including consent withdrawal, please see our Privacy Policy.

Reviewer #1: Yes: Janos G. Pitter MD, PhD

Response: The figure files have been uploaded to PACE.

---

## [Editor Report · Decision Letter 2]

5 Dec 2022

Impact of early detection on cancer curability: a modified Delphi panel study

PONE-D-22-14922R2

Dear Dr. Kim,

We’re pleased to inform you that your manuscript has been judged scientifically suitable for publication and will be formally accepted for publication once it meets all outstanding technical requirements.

Kind regards,

Kush Raj Lohani, MBBS, MS (General Surgery)

Academic Editor

PLOS ONE
---

## [Editor Report · Acceptance letter]

12 Dec 2022

PONE-D-22-14922R2 

Impact of early detection on cancer curability: a modified Delphi panel study 

Dear Dr. Kim:

I'm pleased to inform you that your manuscript has been deemed suitable for publication in PLOS ONE. Congratulations! Your manuscript is now with our production department. 

Kind regards, 

on behalf of

Dr. Kush Raj Lohani 

Academic Editor

PLOS ONE